# Modified Electrodeposited Cobalt Foam Coatings as Sensors for Detection of Free Chlorine in Water

**Modestas Vainoris [1], Natalia Tsyntsaru [1,2] and Henrikas Cesiulis [1,*]**

[1] Department of Physical Chemistry, Vilnius University, Naugarduko str. 24, LT-03225 Vilnius, Lithuania; m.vainoris@gmail.com (M.V.); ashra_nt@yahoo.com (N.T.)

[2] Institute of Applied Physics, Academiei str. 5, MD-2028 Chisinau, Republic of Moldova

[*] Correspondence: henrikas.cesiulis@chf.vu.lt

**Abstract:** Metal foams offer a substantial specific surface area and sturdy frame, which makes them great candidates for various applications such as catalysts, sensors, heat sinks, etc. Cobalt and its various compounds are being considered as a cheaper alternative for precious and rare metal catalysts. The cobalt foams have been electrodeposited under galvanostatic and current pulse modes; the porous surface was created using a dynamic hydrogen bubble template. In order to obtain the highest porosity, four different solutions were tested, as well as a wide current density window (0.6–2.5 A/cm$^2$), in addition many different combinations of pulse durations were applied. The effects of surfactant (isopropanol) on porosity were also investigated. The morphology of obtained foams was examined by SEM coupled with EDS, and XRD spectroscopy. True surface area was estimated based on the values of a double electric layer capacitance that was extracted from EIS data. Cobalt foams were modified using $K_3[Fe(CN)_6]$ solution and cyclic voltammetry to form a cobalt hexacyanoferrate complex on the foam surface. In order to find optimal modification conditions, various potential scan rates and numbers of cycles were tested as well. Free chlorine sensing capabilities were evaluated using chronoamperometry.

**Keywords:** cobalt; electrodeposition; metal foams; cobalt hexacyanoferrate; free chlorine detection

## 1. Introduction

One of the most vital issues for the fast-growing human population is the availability of clean and safe drinking water and sanitation, which was recognized by the United Nations by putting it on their Millennium Development Goals list for the second time [1]. Water disinfection, usually the last step in a water cleaning process, helps to prevent sickness and various diseases [2,3]. Nevertheless, disinfection by-products, that form during the technological process using common disinfectants (ozone, chlorine and chlorine dioxide), can react with leftover organic materials or, bromide and iodide in the water, forming genotoxic and carcinogenic compounds [4,5].

Chlorine being the cheapest option, is usually used for residual disinfection of water and limiting pathogen growth within the water distribution system. Therefore, the monitoring and regulation of chlorine are essential [6–8]. Free chlorine concentration, which is a sum of dissolved chlorine gas, hypochlorous acid and hypochlorite anion, is usually monitored using N,N'-diethyl-p-phenylenediamine and spectrophotometric measurements [9]. In this view, electrochemical methods offer accessible, in situ measurement possibilities, that have good sensitivity and selectivity, and can be reusable and thus cheaper too. There has been quite a lot of interest in the development of diverse electrochemical chlorine sensors and electrochemical techniques [10–16].

Porous metal foams have received a great interest due to their compelling physical and chemical properties, such as high porosity, low density, good electrical, magnetic and mechanical properties, and

make an appealing material for a wide area of applications/devices: catalysis [17–23], fuel cells [24], sensors [25], batteries [26,27] and heat exchangers [28,29]. Recent studies regarding foams based on iron group metals, usually focused on an increase in the surface area of a foam (high surface area catalysts) for hydrogen [19,20,23] or oxygen [21–23] evolution reactions. Commonly, to achieve these aims modified foams are employed. The modified foams have proven to be an attractive alternative for expensive catalysts such as platinum and ruthenium dioxide, allowing to create very active asymmetric electrodes cell for water splitting [23].

Metal foams can be manufactured using different methods including electrochemical ones: either using hard (polymeric or metallic template) or dynamic hydrogen bubble template [18,20,30–32]. Dynamic hydrogen bubble template is based on the use of high current densities, and hydrogen bubbles forming on the substrate surface prevent the deposition of metal, then the metal ions are reduced only in the gaps between gas bubbles, thus leading to the development of the metal foam structure [33]. This method of metal foams production has received much attention, because of the ability to control pore size, deposits density, crystallite size and morphology. Numerous metallic foams have been deposited, however, to the best of our knowledge, no pure cobalt foams have been deposited using a dynamic hydrogen bubble template assisted electrodeposition. Usually galvanostatic, potentiostatic or pulse deposition techniques are used for metal foams production. When the pulse electrodeposition technique is used, one can still form a metallic foam, which offers both micro and nanoscale morphological templating [27,34]. Organic additives and different ions (ammonium ions, BTA, etc.) during metal foams depositions also help to control the porosity and mechanical stability [35–37]. Ligands, which can form weak complexes with cobalt ions and produce hydrogen bubbles during the cathodic reaction, can increase porosity and mechanical stability of cobalt foam's [35].

High surface area and sturdy Co foams could be applied directly or modified creating Prussian blue $Fe_4[Fe(CN)_6]_3$ like complex on the foams surface. Prussian blue and other derivatives with transition metals such as cobalt, nickel, copper etc., have been extensively investigated for many years [10,11,38–43]. Its applications vary from the detection of new molecules [43], ascorbic acid [40], morphine [42], free chlorine in water [10,11], or even used as photoanodes [41].

Thus, the objectives were: To investigate in detail the feasibility of Co foam fabrication, employing a dynamic hydrogen bubble template electrochemical formation method; to reveal the effects of bath chemistry and deposition conditions on the structure and morphology of cobalt foams; to modify foams in $K_3[Fe(CN)_6]_3$ acetate buffer solution and for the first time to apply modified surfaces as sensors for the determination of the concentrations of the free chlorine in the water.

## 2. Materials and Methods

Composition of electrolytes for cobalt foams deposition are presented in Table 1. All of the reagents used were of analytical grade and were used as received without further purification. All of the solutions were prepared using deionized water (DI). Electrodeposition of Co foams was performed at room temperature.

**Table 1.** The chemical compositions of solutions used for deposition of cobalt foams; pH 2.

| Solution No | $CoCl_2$ (mol/L) | $NH_4Cl$ (mol/L) | $CoSO_4$ (mol/L) | $(NH_4)_2SO_4$ (mol/L) | Isopropyl Alcohol (mol/L) |
|---|---|---|---|---|---|
| 1 | 0.2 | 2 | – | – | – |
| 2 | 0.2 | 2 | – | – | 2 |
| 3 | – | – | 0.2 | 1 | – |
| 4 | – | – | 0.2 | 1 | 2 |

### 2.1. Surface Preparation and Deposition of Cobalt Foams

Cobalt foams were electrodeposited using a dynamic hydrogen bubble template method on copper substrate, which was used as a working electrode. The geometrical area of copper foil sheets

was 0.8 cm$^2$. Prior to the electrodeposition, the Cu substrate was mechanically polished, degreased in acetone and then cleaned with DI water in the ultrasonic bath. Before deposition, the native copper oxide layer was removed by dipping the substrate into 2 M H$_2$SO$_4$ solution. In order to improve adhesion of the deposits to the substrate, a Ni seed layer (~10 nm) was deposited from the solution containing 1 M NiCl$_2$ and 2.2 M HCl under galvanostatic mode ($j = -12.5$ mA/cm$^2$) for 1 min.

Two electrode cells were used for the deposition of the cobalt foams, where a circular platinized titanium mesh was used as the counter electrode. The distance between electrodes was fixed at 2.5 cm. The cobalt foams were deposited under galvanostatic or pulse deposition mode. The influence of Cl$^-$ and SO$_4^{2-}$ based electrolytes, of the cathodic current density (0.6–2.5 A/cm$^2$), of the deposition time (20–300 s) on porosity, structure and morphology of cobalt foams was evaluated.

As-deposited cobalt foams were thoroughly rinsed with DI water. Afterwards, coatings were immediately transferred into a beaker with ethanol, in order to minimize contact with the atmosphere and thus avoid oxidation of a highly active surface area of cobalt foams. The true surface area was evaluated using electrochemical impedance spectroscopy (EIS). The scans at different potentials were registered in 0.1 M Na$_2$SO$_4$ solution at four decades of frequencies ($f = 10^3$–0.1 Hz) using a standard three-electrode cell. The saturated Ag/AgCl electrode was used as a reference electrode. The fitting of EIS data was done using ZView software (version 2.8d). EIS measurements were performed at room temperature.

### 2.2. Modification of Co Foams and Detection of Free Chlorine

Cobalt hexacyanoferrate was formed using cyclic voltammetry (CV) in the same three electrodes that were set-up for EIS measurements. Cobalt foams were immersed into 0.05 M ammonium acetate buffer solution with 0.1 M KNO$_3$ and 1.5 mM of K$_3$[Fe(CN)$_6$]; the pH of the chosen buffer solution was fixed at 5.5, adjustments were made using acetic acid. All solutions were freshly prepared. CV measurements were performed at room temperature. In order to find the best conditions for foams modification (i.e., an enhanced sensitivity and longevity of a free chlorine sensor), the influence of different cycling speeds (25, 50, 100 mV/s) and count of cycles was investigated.

Chronoamperometric measurements were performed in order to determine the amount of free chlorine in water. Ca(OCl)$_2$ was used as a source of chlorine. Various amounts of Ca(OCl)$_2$ were dissolved in 0.05 M ammonium acetate buffer, containing 0.1 M KNO$_3$ as a background electrolyte. All the solutions used for chlorine detection were prepared just prior to chronoamperometric measurements. A standard three-electrode system was used during chronoamperometric measurement, with a modified cobalt foam as the working electrode. All of the measurements were performed at room temperature.

### 2.3. Instrumentation

The electrodeposition of Co foams coatings and other electrochemical measurements (cyclic voltammetry, chronoamperometry, EIS etc.) were performed using programmable potentiostat/galvanostat AUTOLAB PGSTAT302N (Methohm, Ultrech, The Netherlands), controlled using Nova 1.10 software. Cobalt foams morphology was characterized using scanning electron microscopes (Hitachi's Tabletop Microscope TM3000, and Hitachi's SU-70, Tokyo, Japan). Energy-dispersive X-ray Spectroscopy was used to determine the elemental surface composition of as-deposited and modified Co foams. X-ray diffractometry (XRD, Rigaku Miniflex II, Tokyo, Japan, $\lambda = 1.5418$ Å/Cu K$\alpha$, $2\theta = 20°{\rightarrow}100°$, 10°/min) was used to evaluate the phase composition of obtained cobalt foams. Porosity of cobalt deposits were calculated using SEM images and Fiji image processing software (version 1.52n), considering the pores as voids and subtracting their area from the total surface area of a SEM image.

## 3. Results and Discussion

### 3.1. The Influence of Electrolyte Composition and Operating Conditions on Co Coatings Deposition

As it was shown by DoHwan Nam et al. [37] the ammonium ions played an important role in copper foams deposition, whilst using a dynamic hydrogen bubble template method. The ammonium ions in water-based solutions were in the equilibrium with $NH_3$:

$$NH_3 + H^+ \Leftrightarrow NH_4^+ \tag{1}$$

In our case, the solution used for depositions was acidic (pH 2), thus the equilibrium shifted towards $NH_4^+$. Ammonium ions can act as a ligand forming complexes with cobalt ions, but may also adsorb onto the surface of a substrate during deposition. Reduction of the adsorbed ammonium ions on the cathode leads to a decrease of the cathodic current efficiency, but also acts as an additional hydrogen source (Equation (2)), which in turn influences the porosity of the obtained coatings [44]:

$$NH_4^+ + e^- \rightarrow 0.5H_2 + NH_3 \tag{2}$$

In order to reveal the importance of ammonium ions, whilst depositing cobalt foams coatings, the ammonium-free electrolytes that contained only 0.2 M $CoCl_2$ or 0.2 M $CoSO_4$ were used. The galvanostatic deposition at various cathodic current densities (0.6–4.8 A/cm$^2$) and deposition times (10–180 s) were studied, but a foam-like structure was not obtained (Figure 1). The coatings were extremely uneven using both sulfate- and chloride-based electrolytes, and the jet of formed hydrogen bubbles removed most of the reduced metal from the substrate surface. Nevertheless, as it is depicted in SEM images the coatings obtained from chloride-based electrolyte had a better coverage (Figure 1a) than coatings obtained from the sulfate bath (Figure 1b) that were uneven and had micro-agglomerates on the surface. Both time and current density did not positively affect the formation of cobalt foams from the electrolytic baths.

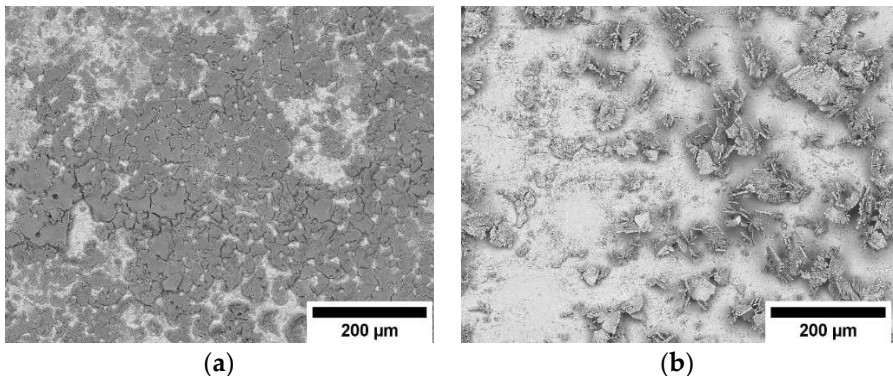

(**a**)       (**b**)

**Figure 1.** SEM images of Co coatings deposited under galvanostatic conditions, at cathodic current density $j$ = 2.5 A/cm$^2$, deposition time $t$ = 60 s. The composition of solutions: (**a**) 0.2 M $CoCl_2$; (**b**) 0.2 M $CoSO_4$.

The addition of ammonium ions (Figure 2) to the solutions resulted in the deposition of Co coatings riddled with cylindrically shaped pores of various sizes, but often displaying numerous defects, caused by hydrogen evolution. Coatings electrodeposited from the chloride-based solution (solution 1) had larger and irregular pores (diameter of pores was varied from 5 to 100 μm) in comparison to the sulfate-based solution, where pores size were much more uniform (diameter 5–20 μm solution 3). This fact can be explained by the higher capability of the ammonium sulfate in the suppression of hydrogen bubble coalescence compared to the ammonium chloride solutions [45].

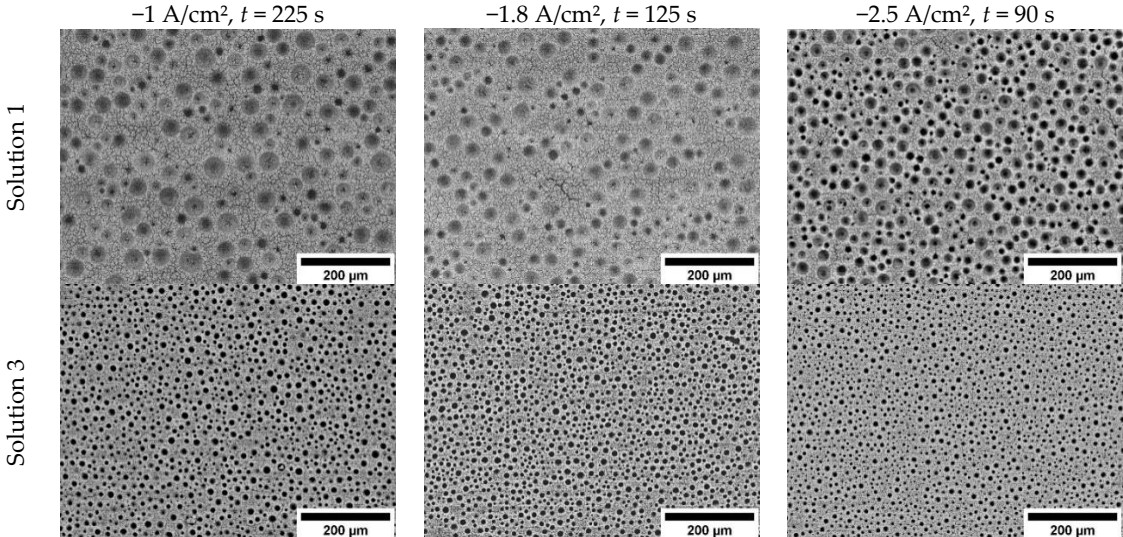

**Figure 2.** Scanning Electron Microscopy images of cobalt coatings obtained under a galvanostatic mode at various deposition conditions, but the same value of charge passed ($q$ = 180 C).

The influence of the cathodic current density on the foam's formation revealed the following aspects: From the chloride-based solution, the foam-like structure was obtained only at cathodic current densities >1.8 A/cm$^2$ (Figure 2). The porosity of coatings obtained from solution 1 alternated dependently on the cathodic current density ($q$ = 360 C), namely: 19% at 1 A/cm$^2$, 23.3% at 1.8 A/cm$^2$ and 21.7% at 2.5 A/cm$^2$. The increase in porosity from the chloride-based solution at higher cathodic current densities could be explained by the increase in the rate of a secondary reaction–hydrogen evolution reaction. As it was mentioned above, the ammonium sulfate was approximately three times better at suppressing hydrogen bubbles coalescence, and thus, the diameter of pores usually does not exceed 15 μm for foams obtained from a sulfate-based solution. Hence, the overall porosity of such foams is much lower than the ones obtained from a chloride-based solution: At −2.5 A/cm$^2$ porosity is ca 8.2%.

In order to evaluate the role of a surfactant on the formation of Co foams, the isopropyl alcohol was added (solutions 2 and 4), and the porosity and morphology of the growing coatings was evaluated. The isopropanol was chosen as an efficient agent to reduce the surface tension. Thus, the hydrogen bubbles formed during deposition, were able to easily detach from the surface. As it is apparent from the SEM pictures (Figure 3), the porosity of the deposits increased substantially even at comparatively low current densities of 1 A/cm$^2$. However, the radius of pores did not seem to be affected by the reduction of the solutions surface tension. This could be linked to the fact, that the influence of ammonium ions on hydrogen bubbles is much higher than that of isopropyl alcohol on the reduction of surface tension. The porosity of the foams obtained from the sulfate-based solutions also increased substantially with the addition of isopropyl alcohol. Thus, at a cathodic current density of 2.5 A/cm$^2$ the development of tridimensional cobalt foam structure can be noticed, and it is formed from interconnected cylindrical pores (Figure 3). It this case the estimation of the porosity using SEM images was rather difficult due to many interconnected pores.

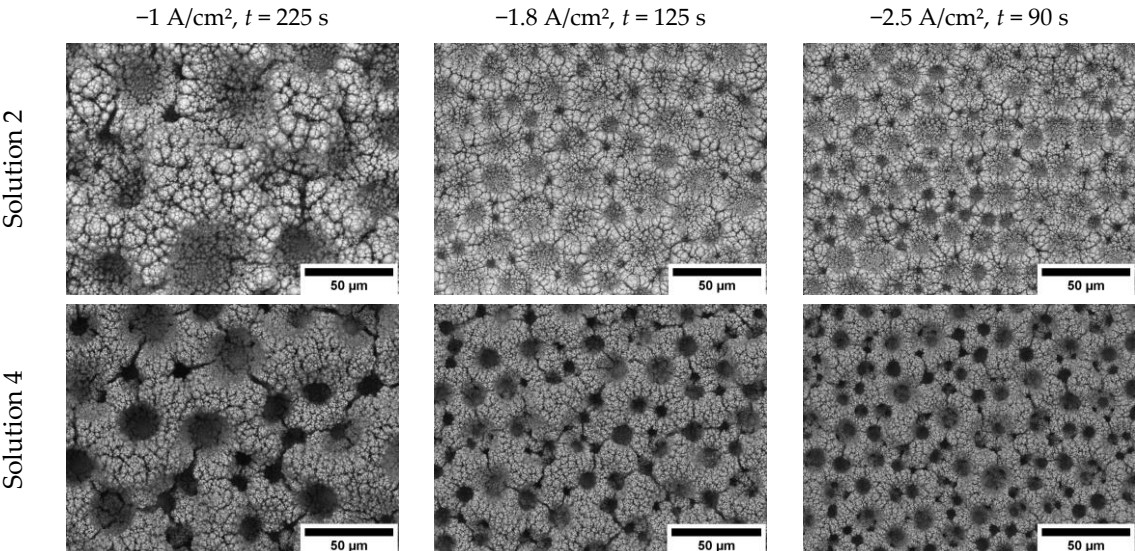

**Figure 3.** Scanning Electron Microscopy images of cobalt coatings obtained under a galvanostatic mode at various deposition conditions, but the same value of charge passed ($q$ = 180 C), from solutions containing isopropyl alcohol.

The two main reactions occurring during depositions using such high current densities were reduction of cobalt ions, and reduction of water. The use of a dynamic hydrogen bubble template takes advantage of hydrogen evolution, forcing the metal ions to be reduced only in-between the hydrogen bubbles. The decrease in surface tension and easier detachment of formed bubbles resulted in an increase in the area for metal ions reduction, however cobalt is a very good catalyst for water reduction reaction, and isopropanol addition makes the water reduction reaction the dominant reaction by a huge margin. This was evident from current efficiency (CE) data (Figure 4), which was calculated using Faraday's law. The hollow figures display solutions with the isopropanol.

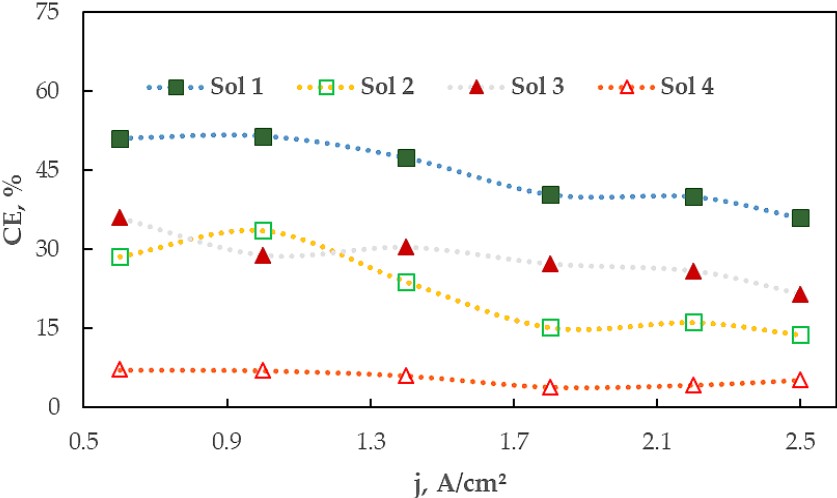

**Figure 4.** Influence of the cathodic current density on the current efficiency (CE) of Co coatings obtained under a galvanostatic mode and $q$ = 180 C.

There were three clear trends for CE dependence on applied current density which were seen: (1) Increasing cathodic current density decreased the amount of deposited metal; (2) the use of sulfate based electrolytes (solutions 3 and 4) led to less deposited metal compared to chloride-based electrolytes (solutions 1 and 2); (3) the isopropyl alcohol significantly lowered CE. The first phenomenon can be explained by an increased overpotential, which in turn increased the rates of both reactions (Co and

water reduction). The cobalt ions reduction was controlled by the diffusion rate, which affected the quantity of deposited metal with increased current density. The second trend can be explained by the significantly different capabilities of hydrogen bubbles coalescence suppression. The smaller bubbles cover the surface, encumbering the diffusion of metal ions around them. The third phenomenon is caused by significant reduction of surface tension, when using isopropyl alcohol. With reduced surface tension, hydrogen bubbles detach faster, and since the reduction reaction is diffusion controlled, so HER dominates even more, hence the increase in porosity and the formation of cobalt foams.

The use of such high current densities could affect also the preferred crystallographic orientation of Co. XRD spectra (Figure 5) recorded for cobalt foams electrodeposited from solution 4 by applying various current densities showed a clear face centered cubic (fcc) structure. Usually for electrodeposited cobalt coatings the hexagonal close packing (hcp) structure is indexed [46]. The fcc structure of Co with two most intense peaks (1 1 1) and (2 0 0) are commonly obtained applying other methods [47]. Nevertheless, electrodeposited Co using high current densities formed crystalline deposits with most preferred plains (2 0 0) and (2 2 0), while the intensity of (1 1 1) was comparable to that of the (3 1 1) plane, or even less for higher cathodic current densities. That outcome could be linked to the distortion created by the evolution of hydrogen bubbles.

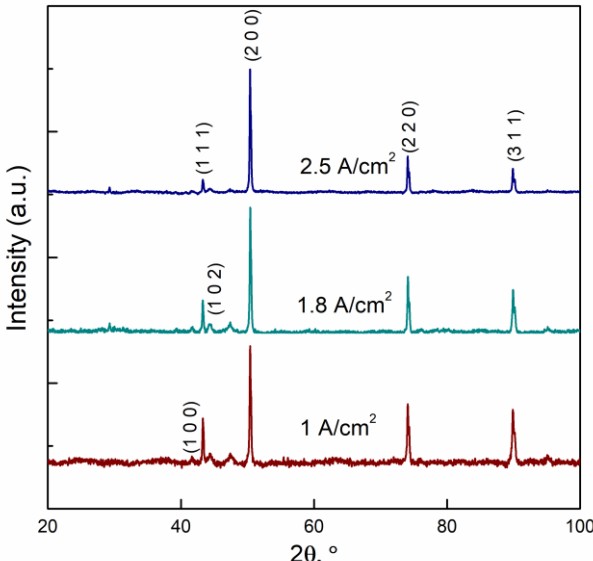

**Figure 5.** XRD pattern of Co foams electrodeposited from solution 4. Peaks were analyzed according to JCPDS cards No. 01-071-4238 and 01-077-7453.

### 3.2. The Mechanism of Co Foams Depositon and Determination of True Surface Area

As it was mentioned earlier when using the dynamic hydrogen bubble template, high current densities were applied, and deposition occurred in-between hydrogen bubbles, whilst being diffusion controlled. SEM top (Figure 3) and cross-sectional (Figure 6) images of cobalt foams, coupled with EDS data of the surface, revealed that after the initial formation of crystallization centers, intensive growth of fern-like cobalt agglomerates occurred, in such a way forming cobalt foam, leaving cylindrically-shaped pores. In the course of deposition, the radius (size) of the pores increased. Such an increase can be explained by the increase in the HER rate, caused by the exposed high surface area of the cobalt deposits, since it is well-known that cobalt is a good catalyst for the HER reaction [48–50].

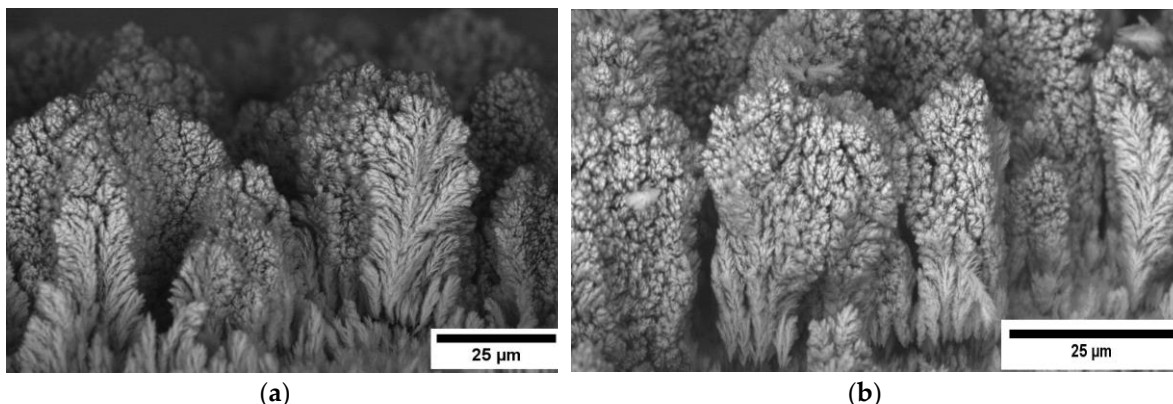

(**a**)                                                    (**b**)

**Figure 6.** SEM images of cobalt foams cross-sections deposited under galvanostatic mode, from solution 4: (**a**) $j$ = 1 A/cm$^2$, $t$ = 225 s; (**b**) $j$ = 2.5 A/cm$^2$, $t$ = 90 s.

With huge differences in porosities and CE, the true surface area was estimated by electrochemical impedance spectroscopy under the assumption that the capacitance of the double electric layer of the same metal depended on the real surface area. To ensure the reliable characterization, the EIS measurements were performed across a wide range of frequencies (10 kHz to 0.1 Hz), at selected cathodic potentials– −0.8, −1 and −1.2 V (vs Ag/AgCl). An example of EIS spectra is presented in Figure 7 recorded using cobalt foams obtained from Solution 4 at different current densities. Fitting of impedance data was done using an equivalent scheme containing two pairs of constant phase elements in parallel with resistances (inset in Figure 7). We could see only one clear capacitive behavior, however since the electrodes were very porous and EIS was measured during HER, the physical meaning had to have an equivalent electric circuit (EEC) containing at least two capacitors. In our case, the EEC shown in Figure 7 showed a best fit for the obtained EIS spectra (minimal Chi-squared and elements values errors). First capacitance could be attributed to: (1) Porosity of the electrode, (2) double layer formation, or (3) diffusion limitations. According to Mulder et al., at highly contorted surfaces (3D) one can expect to obtain CPE n values of around 0.5 [51]. It was shown that when the first (high frequency) semicircles radius was potential independent, it was related to porosity and the shape of pores [52,53]. However, in our case the high frequency semicircle capacitance changed, hence it could not be purely porosity related. Also, during measurement evolving hydrogen bubbles might have blocked up pores and restricted further hydrogen evolution, then in impedance spectra it would seem like diffusion limitation.

Moreover, Łosiewicz et al. [54] showed that charge-up of the double layer of porous electrodes usually occurred not uniformly (there was frequency dependence). Hence the second semicircle (low frequency) could then be attributed to modeling of a charge transfer resistance process and differential charge-up of the double-layer. It was also shown [54] that double layer capacitance reduced with an increase in the overpotential in HER, which showed similar behavior to the low-frequency semicircle values like in our case. Nevertheless, the first semicircle probably was related to the combination of double layer partial charge-up process, porosity of electrode and maybe even diffusion processes, but separation of given processes in our case was quite difficult.

The second semicircle, located in the low frequencies region that also changed with potential was attributed to double layer capacitance and adsorption of hydrogen on the surface of cobalt foams. Both of these processes were potential dependent. All our efforts to separate the two processes were fruitless.

All the calculations and comparisons were made from the data obtained at −1.0 V vs Ag/AgCl. The equivalent circuit model (Figure 7) represented our best efforts, as it can be seen that it fit quite well with the measured data. The Chi-squared values were usually in the range from $10^{-4}$ to $10^{-5}$, which was acceptable for porous electrodes. Some examples are shown in Table 2. The second constant phase element value was recalculated into true capacitance values using Mansfeld's procedure. The calculated capacitances were much higher than the ones obtained for flat metallic cobalt, which did

not exceed 100–200 µF/cm$^2$. It indicated that the obtained porous electrodes had 100–300 times larger actual surface area than the flat surface.

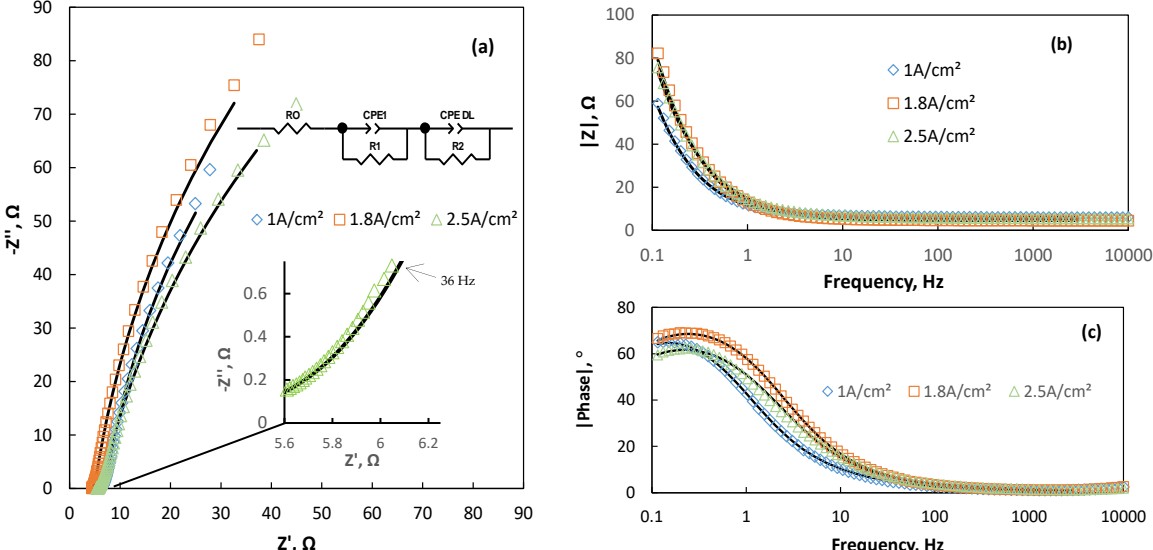

**Figure 7.** EIS data of cobalt foams measured in 0.1 M Na$_2$SO$_4$ at −1.0 V vs Ag/AgCl, foams deposited from solution 4, *q* = 180 C, EIS data represented in: (**a**) Nyquist's coordinates, (**b**) Bode modulus, and (**c**) Bode phase modulus coordinates. The equivalent electric circuit used for fitting is presented in the insert; points–experimental data, solid lines–results of fitting to the shown equivalent electric circuit.

**Table 2.** Values of elements of the equivalent electric circuit fitted EIS data obtained in 0.1 M Na$_2$SO$_4$ at −1.0V vs Ag/AgCl of cobalt foams deposited at *j* = 2.5 A/cm$^2$, *t* = 90 s.

| Solution Used for Co-Foam Deposition | $R_0$, Ω | $CPE_1$, Fs$^{n-1}$ | $n$ | $R_1$, Ω | $CPE_{DL}$, Fs$^{n-1}$ | $n$ | $R_2$, Ω |
|---|---|---|---|---|---|---|---|
| 1 | 3.668 ± 0.50% | 0.1131 ± 1.7% | | 0.7317 ± 3.5% | 0.0342 ± 0.91% | 0.9321 ± 0.54% | 133.7 ± 5.5% |
| 2 | 4.081 ± 0.27% | 0.0499 ± 1.6% | 0.5 | 0.7652 ± 1.4% | 0.0112 ± 0.37% | 0.8844 ± 0.17% | 349.7 ± 1.8% |
| 3 | 4.872 ± 0.43% | 0.0468 ± 1.9% | | 0.878 ± 2.1% | 0.028 ± 0.82% | 0.8313 ± 0.48% | 508.8 ± 10.7% |
| 4 | 5.427 ± 0.53% | 0.046681 ± 2.9% | | 0.80871 ± 2.3% | 0.01736 ± 0.94% | 0.85157 ± 0.45% | 311.9 ± 9.1% |
| Pure Co | 8.442 ± 0.17% | – | – | – | 0.000181 ± 0.402% | 0.905 ± 0.0868% | 1458 ± 0.79% |

As it might seem, the second CPE element values (double layer and hydrogen adsorption capacitance), obtained using chloride-based solutions got the highest double layer capacitance, thus the highest surface area of cobalt foam. Nevertheless, after recalculations of CPE to true capacitance and further calculations of capacitance per gram of cobalt foam (Figure 8), the results changed completely–the highest surface area to mass ratio was obtained for foam electrodeposited from solution 4. This could be explained by higher efficiency of ammonium sulfate in hydrogen coalescence suppression, and lowered surface tension using isopropyl alcohol. These factors allowed the growing coating to form a tridimensional porous structure, with interconnected pores. All of the other foams were quite similar in surface area exposed per unit of mass; however, the ones obtained from solution 4, were three or more times superior.

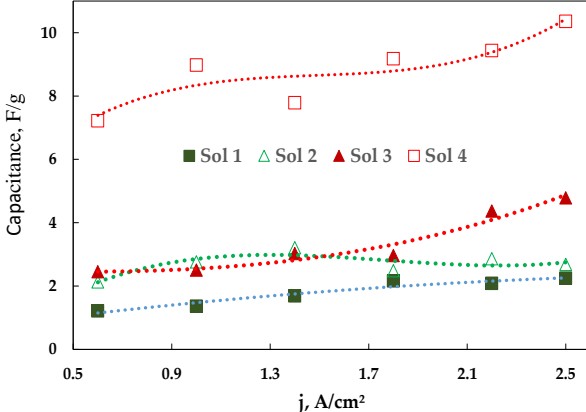

**Figure 8.** Influence of cathodic current density on the capacitance per gram of cobalt foams in the investigated solutions.

Cobalt foams obtained using pulse deposition, showed better CEs across the board, sometimes reaching up to 50% in efficiency, hence the obtained foams were thicker. Nevertheless EIS analysis of these foams showed that surface area to mass ratio was in all cases lower, than the one obtained using galvanostatic deposition conditions. Further research was done using only cobalt foams obtained at 2.5 A/cm$^2$ using solution 4 ($q$ = 180 C).

### 3.3. Modification of Cobalt Foams and Detection of Free Chlorine

As mentioned earlier, modification of cobalt foams was with ferrocyanide, which was done using cyclic voltammetry. The measurements were performed in acetic acid buffer solution. Such a buffer was chosen because of highly active and substantial cobalt foams surface area. Other buffer solutions such as phosphate or citrate were tested, however either the foam reacted with the anions in the buffer (phosphate), or formed complexes at the surface (citrate case), diminishing the modification capabilities. The potassium nitrate used as background electrolyte, was shown to have second best ion permeability in various metal HCFs structures, ensuring good conductivity and electrons exchange [10,38]. The chosen potential window for modification was −0.8–0 V vs Ag/AgCl.

The potential in CV scans was first moved to the anodic side, in order to dissolve some of the foam, and afterwards to cathodic, to form cobalt ferrocyanide complex on the surface of the foams. Comparatively high scan rates were used to form cobalt ferrocyanide, so the damage to high surface area cobalt foams would be minimized. Additionally, no clear peaks could be seen on the voltammetric curve, since the scan speed used was quite high, the Co foams surface was very porous, and the response might have been too slow to be detected in such a system. A high scan rate was selected in order to minimize the damage of the cobalt foams surface, trying to keep the highest possible surface area intact, but still covering the whole surface of foam with Co HCF complex. With no clear oxidation or reduction peaks obtained during CV scans, a formation of cobalt ferrocyanide on the surface of cobalt foams had to be done externally. For that reason, EDS measurements were performed to ensure successful formation of cobalt ferrocyanide complex (Figure 9). The EDS picture represents a typical modified cobalt foam. It can be seen that the whole surface was quite uniformly covered with Fe and C compound, proving successful formation of Co HCF complex on the surface of the Co foam.

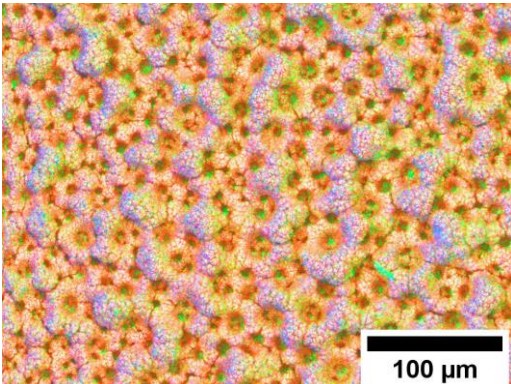

**Figure 9.** Image of EDS mapping of modified Co foam, using CV at 100 mV/s scan rate and after 100 cycles. Red color represents cobalt, green–iron, blue–carbon.

Free chlorine concentration, which is a sum of dissolved chlorine gas, hypochlorous acid and hypochlorite anion, was detected using acetic acid buffer solution whilst measuring the amperometric response of our system. Prior to the testing the purity of $Ca(OCl)_2$, which was used as a source of free chlorine and checked using the standard DPD method. Determined purity of calcium hypochlorite was 64%. The chosen pH value for chlorine detection measurements was −5.5 and was done so for two reasons: (1) Trying to simulate real tap water pH range, which was usually 5 < pH < 8, (2) all of the compounds of free chlorine exist in the solution at such pH values. However, most of the chlorine exists in the form of HClO, since the $pK_a$ of reaction (3) was 7.48 at 25 °C.

$$HClO \rightarrow ClO^- + H^+ \tag{3}$$

Hypochlorous acid reduction occurs in two steps (reaction 4 and 5), which according to Cheng et al. If the pH is above 3 is irreversible [55], making it possible to fully reduce and hydrolyze free chlorine in water to chloride anions.

$$HClO + H^+ + e^- \rightarrow 0.5Cl_2 + H_2O \tag{4}$$

$$Cl_2 + 0.5H_2O \rightarrow HClO + H^+ + Cl^- \tag{5}$$

All of the free chlorine-containing compounds were reduced electrochemically very easily, so in order to minimize the damage of high surface area cobalt foams, ensuring longevity of the sensor, a plethora of potentials were tested. The best results were obtained using −0.45 V vs Ag/AgCl potential with the cobalt foams modified for 40 cycles at 100 mV/s speed. A calibration curve and typical chronoamperometric measurements are shown in Figure 10. The linearity of the curve was slightly distorted probably by oxygen reduction that was dissolved in testing solutions. Nevertheless, the increase of cathodic current with the addition of very small amounts of chlorine into water, proved very high sensitivity for our prepared sensor. The standard deviation of the blank solution was calculated using data from 10 chronoamperometric measurements. The calculated limit of blank (LOB = 1.65σ) was 3.06 ppb. The obtained limit of detection (LOD = 3σ) was 5.57 ppb and limit of quantification (LOQ = 10σ) was 18.86 ppb. Such results were acceptable, since the usual concentration of residual chlorine in tap water was somewhere between 0.2 and 1 ppm.

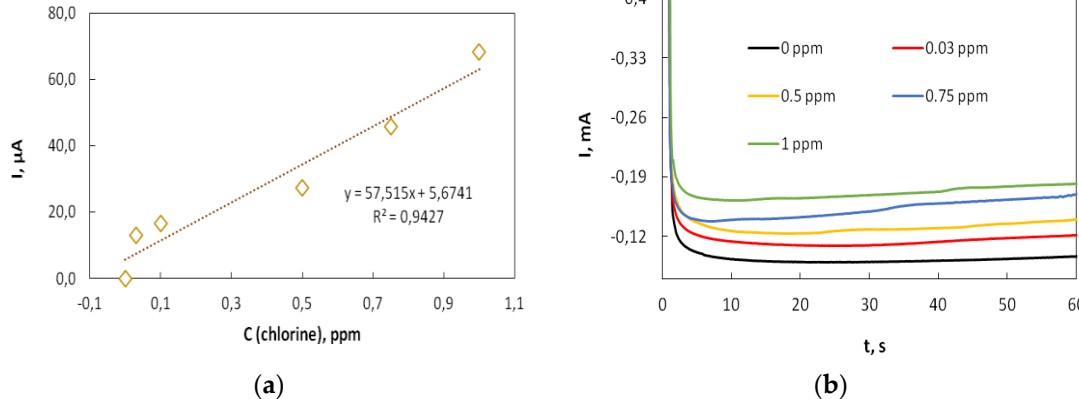

**Figure 10.** Calibration curve for sensors (**a**) and typical chronoamperometric curves for free chlorine detection in acetic acid buffer (**b**).

## 4. Conclusions

In the present work, cobalt metal foams were successfully deposited using a dynamic hydrogen bubble template method on the copper substrate. The true surface area of cobalt foams was estimated using the EIS technique. It was determined that the highest surface area cobalt foams were electrodeposited using a solution containing 0.2 M $CoSO_4$, 1 M $(NH_4)_2SO_4$ and 2 M isopropyl alcohol at a cathodic current density of 2.5 A/cm$^2$. In this case, there was synergy between the ammonium sulfate bubble suppression effect, and solutions surface tensions reduction, which allowed for the formation of highly porous 3D structured cobalt foams. The cobalt foams surface was modified with Co hexacyanoferrate, and such modified foams have been tested as sensors for the detection of free chlorine in water. A linear range from 5.6 ppb to 1 ppm was shown. It was demonstrated that such a sensor can be good and a cheaper alternative to noble metal sensors currently used for the detection of the concentration of residual chlorine in water.

**Author Contributions:** Investigation, M.V.; Methodology, M.V.; Supervision, H.C. and N.T.; Visualization, M.V.; Writing–original draft preparation, M.V.; Writing–review and editing, M.V., H.C. and N.T.

**Funding:** This research received funding from Horizon 2020 research and innovation program under MSCA-RISE-2017 (No. 778357) and from the Research Lithuanian Council project (No 09.3.3-LMT-K-712-08-0003).

**Conflicts of Interest:** The authors declare no conflict of interest.

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
