# Peer review of "Modified Electrodeposited Cobalt Foam Coatings as Sensors for Detection of Free Chlorine in Water"

_coatings, doi:10.3390/coatings9050306_

Round 1
Reviewer 1 Report
The manuscript "Modified electrodeposited cobalt foam coatings as sensors for detection of free chlorine in water” by M. Vainoris et al. reports the electrochemical synthesis of cobalt foams as efficient sensors for free chlorine in water by means of dynamic hydrogen bubble template. The manuscript is well structured with clear messages and nice readable graphs. The results are helpful to other researchers in this field. However, there are various points that must be re-considered before publication in any case:
In introduction section, the authors should be addressed recent papers and highlight their hypothesis, new concepts and innovations briefly. Novelty must be highlighted.
In table 1, pH values should be provided.
In experimental section the working temperature should be indicated.
Further characterization such as BET… can be provided to improve the quality of this work.
Author Response
Thank you for the comments. Please find below attached word file with our responses.

Reviewer 2 Report
Need to improve English writing. Also, EIS measurements need improvements regarding data representation. Should have to mention errors/ standard deviations along with values. Also, specify the LOD and LOQ formula with a standard deviation of the blank solution.

Author Response
Thank you for the comments and questions. Please find below attached word document with our responses.

Round 2
Reviewer 2 Report
Great work.